# BeliefFormer: Belief-Attention in Transformer

## Abstract

In this paper, we consider modifying the attention layer in Transformer to improve its generalization performance. Conceptually speaking, the standard attention layer takes the softmax-based weighted summation of V vectors as the residual signal (with a linear mapping for post-processing) when performing the skip-connection operation. Inspired by distributed optimization, we propose to first perform an orthogonal projection of the softmax-based weighted summation of V vectors with respect to the original V vectors and then take the perpendicular component instead as the residual signal (with a linear mapping for post-processing) when performing the skip-connection operation. By doing so, the token vectors are modified relatively more along their tangent directions compared to their magnitudes. Intuitively speaking, the perpendicular component reflects a belief about the discrepancy between the weighted summation of V vectors and the V vectors themselves. We refer to the newly modified layer and the overall architecture as the belief-attention and the BeliefFormer, respectively. To further improve performance, we also design a variant of belief-attention by incorporating both the per attention-head based and global orthogonal projections, referred to as belief-attention*. Experimental results show that the two new variants of attention layer in Transformers lead to better performance than the standard attention for image classification over ImageNet and natural language processing when training nano-GPT2 and Llama.

## 1 Introduction

In recent years, Transformers (Vaswani et al., 2017) have made significant advances across a range of data analysis fields, including natural language processing (NLP) (Achiam et al., 2023; Touvron et al., 2023), computer vision (Dosovitskiy et al., 2021), image generation and editing (Peebles & Xie, 2023; Hatamizadeh et al., 2024; Zhang et al., 2023), and audio processing (Latif et al., 2023). A fundamental component of transformers is the attention layer, which enables the model to capture long-range dependencies within a sequence of tokens. This mechanism works by computing a weighted summation of the value (V) vectors based on the similarity between query ($Q$) and key ($K$) vectors, determined via a softmax function. Conceptually, the attention operation allows each token to aggregate relevant information from all other tokens. Following the attention layer, a feedforward network (FFN) processes each token independently, which can be interpreted as local information fusion. Recent large language models (LLMs) exploit a so-called mixture of experts (MoE) as an extension of basic FFN to improve the performance, where at the inference stage, only certain percentage of weights in the FFN layer are activated depending on the particular input.

One prominent research direction focuses on reducing the quadratic computational complexity inherent in the standard attention layer when processing long token sequences. Various simplified attention schemes have been proposed, which include, for example, LinFormer (Wang et al., 2020), LongFormer (Beltagy et al., 2020), ReFormer (Kitaev et al., 2020), FlashAttention (Dao, 2023), RingAttention (Liu et al., 2023), BurstAttention (Sun et al., 2023). FlashAttention is being widely used in practical situations as it reduces the computational complexity considerably without introducing any approximation in the standard attention layer.

In this work, we attempt to modify the attention layer to improve the generalization performance of Transformers. To do so, we draw inspiration from distributed optimization. From a high-level point of view, the attention-FFN framework in Transformers exhibits a certain similarity to the framework of distributed optimization over an undirected graph. In general, a typical distributed optimization algorithm (see (Zhang & Heusdens, 2018; Boyd et al., 2011)) iteratively alternates between information-aggregation and information-fusion operations until all nodes in the graph reach consensus. Typical algorithms include alternating direction method of multipliers (ADMM)

(Boyd et al., 2011) and primal-dual method of multipliers (PDMM) (Zhang & Heusdens, 2018). Considering PDMM as an example, it was primarily designed to solve the following separable convex optimisation problem over an undirected graph $G = (\mathcal{V}, \mathcal{E})$ representing a pear-to-pear (P2P) network from practice:

$$\min \sum_{i \in \mathcal{V}} f_i(x_i) \text{ subject to } A_{ij}x_i + A_{ji}x_j = b_{ij}, \quad (i,j) \in \mathcal{E}, \tag{1}$$

where each node $i$ carries a local objective function $f_i(\cdot) : \mathbb{R}^{d_i} \to \mathbb{R}$ and each edge $(i,j)$ carries a linear equality constraint as specified by the constant $(A_{ij}, A_{ji}, b_{ij}) \in (\mathbb{R}^{d_{ij} \times d_i}, \mathbb{R}^{d_{ij} \times d_j}, \mathbb{R}^{d_{ij}})$. As will be discussed in detail in Section 3, at each iteration of quadratic-approximation based PDMM (QA-PDMM), each node in the network performs information aggregation from neighbours (corresponding to attention in Transformer) and local information fusion (corresponding to FFN). One key property of QA-PDMM is that its update expression utilizes the consensus discrepancy in terms of the residual error of the linear edge-constraints in (1), which is essential to make the algorithm converge.

We consider extending the standard attention by drawing inspiration from QA-PDMM. In particular, we propose to first perform orthogonal projection of the softmax-based weighted summation of V vectors with respect to their respective original V vectors. The perpendicular component (see Fig. 2 for demonstration) is then taken as the residual signal, and is further processed by a linear mapping for post-processing in preparation for skip-connection. The above newly designed attention, referred to as belief-attention, encourages updates to the token vectors more in their tangent directions and less in their magnitudes. The overall Transformer architecture with belief-attention is referred to as BeliefFormer. In brief, we make three contributions in the paper:

- Belief-attention is proposed as an extension of attention by taking the perpendicular component after orthogonal projection as the residual signal. The perpendicular component provides a belief about the discrepancy between the softmax-based weighted summation of V vectors and V vectors themselves.

- A variant of belief-attention is also proposed by combining both the per attention-head based and global orthogonal projections for capturing richer information, referred to as belief-attention$^*$.

- Experimental results on training nano-GPT2 and Llama for natural language processing (NLP), and training ViTs over Imagenet, CIFAR10 and CIAR100, show that usage of belief-attention and/or belief-attention$^*$ demonstrate considerable improvement in validation performance.

## 2 RELATED WORKS

We note that in this work, the proposed belief-attention and its variant essentially make use of the angle information between two vectors for performance improvement in Transformer. In the literature, the work Karpukhin et al. (2020) applied the dot-product of two vectors to measure their similarities for NLP. In Khattab & Zaharia (2020); Steck & Ekanadham (2024); Zhang (2024), cosine similarity of two vectors is exploited for either NLP or design of optimizers for training deep neural networks (DNNs). Another related work is Bachlechner et al. (2020), which proposed a so-called ReZero to effectively train a particular type of DNNs with skip connections such as ResNet He et al. (2015). The basic idea of ReZero is to introduce trainable parameters in front of the residual signals and initialize them to be zero to make the training more effective.

## 3 BRIEF REVIEW OF QA-PDMM

To facilitate node-oriented distributed optimization of (1) over the graph $G$, QA-PDMM introduces two Lagrangian multipliers $\lambda_{i|j}$ and $\lambda_{j|i}$ for the linear constraint over the edge $(i,j) \in \mathcal{E}$. Let $\mathcal{N}_i$ denote the set of neighbors for node $i$. We summarize the update expression in a proposition below:

**Proposition 1.** *Suppose at iteration $k = 0$, the primal variables and their Lagrangian multipliers are initialized to be $\{x_i^0\}_{i \in \mathcal{V}}$ and $\{\lambda_{i|j}^0 | j \in \mathcal{N}_i, i \in \mathcal{V}\}$, respectively. We further introduce the initialization for the primal variables $\{x_i^{-\frac{1}{2}} = x_i^0\}_{i \in \mathcal{V}}$ at iteration $k = -\frac{1}{2}$. At the $k$th ($k \geq 1$) iteration, each new update $(x_i^{k+\frac{1}{2}}, x_i^{k+1})$ and its associated Lagrangian multipliers $\{\lambda_{i|j}^{k+1} | j \in \mathcal{N}_i\}$*

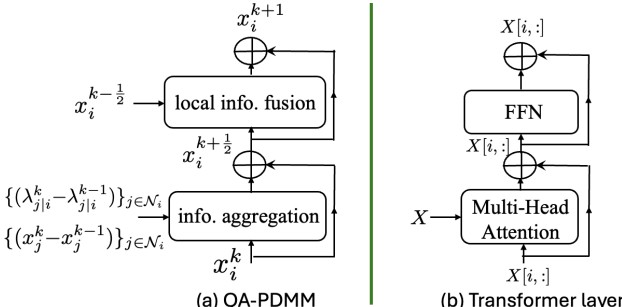

Figure 1: Structural demonstration of QA-PDMM and Transformer layer. The Multi-Head Attention layer utilizes all the tokens in $X$ to update the $i$th token $X[i, :]$ (check (6)-(9) for details).

*at node $i$ are computed as*

$$x_i^{k+\frac{1}{2}} = \overbrace{x_i^k}^{\text{skip-connection}} + B_i^{-1}\Big( \underbrace{\sum_{j \in \mathcal{N}_i} A_{ij}^T(\lambda_{j|i}^k - \lambda_{j|i}^{k-1}) - \rho A_{ji}(x_j^k - x_j^{k-1}))\Big)}_{\text{info. aggregation}} \qquad i \in \mathcal{V} \quad (2)$$

$$x_i^{k+1} = \underbrace{x_i^{k+\frac{1}{2}} + B_i^{-1}(\eta(x_i^{k+\frac{1}{2}} - x_i^{k-\frac{1}{2}}) - (\nabla f_i(x_i^{k+\frac{1}{2}}) - \nabla f_i(x_i^{k-\frac{1}{2}}))}_{\text{skip-connection}} \qquad i \in \mathcal{V} \quad (3)$$

$$\lambda_{i|j}^{k+1} = \underbrace{\lambda_{j|i}^k + \rho(b_{ij} - A_{ji}x_j^k - A_{ij}x_i^{k+1})}_{\text{accumulation of residual signals}} \qquad \forall j \in \mathcal{N}_i, i \in \mathcal{V}, \quad (4)$$

*where $B_i = (\eta I + \rho \sum_{i \in \mathcal{N}_i} A_{ij}^T A_{ij})$, and the stepsizes $\eta, \rho > 0$. See Appendix B for derivation of QA-PDMM from PDMM. The stepsize $\eta$ should be chosen based on the functional property of $f_i$.*

By inspection of (2)-(3), it is seen that $x_i^{k+\frac{1}{2}}$ is computed by first aggregating information from neighbors and then performing the skip-connection. On the other hand, $x_i^{k+1}$ is updated by first performing local information fusion with $(x_i^{k+\frac{1}{2}}, x_i^{k-\frac{1}{2}}, \nabla f_i(x_i^{k+\frac{1}{2}}), \nabla f_i(x_i^{k-\frac{1}{2}}))$ and then applying the skip-connection again. As demonstrated in Fig. 1, the update expressions of QA-DPMM indeed share a great similarity with the structure of a standard Transformer layer from a high-level viewpoint.

Next we study the Lagrangian multiplier $\lambda_{j|i}^k$ being explored in the computation of $x_i^{k+\frac{1}{2}}$. It is not difficult to conclude from (4) that $\lambda_{j|i}^k$ can be represented as a summation of the historical residual errors of the linear equality constraint for edge $(i, j) \in \mathcal{E}$. For the case of $k$ being even, $\lambda_{j|i}^k$ can be represented as

$$\lambda_{j|i}^k = \lambda_{j|i}^0 + \rho \sum_{m=1}^{k/2}(b_{ij} - A_{ji}x_j^{2m-2} - A_{ij}x_i^{2m-1}) + \rho \sum_{m=1}^{k/2}(b_{ij} - A_{ji}x_j^{2m-1} - A_{ij}x_i^{2m}). \quad (5)$$

We take each residual error in (5) as the measurement of the consensus discrepancy between the pair of nodes $(i, j)$. As a result, $\lambda_{j|i}^k$ is computed by accumulating the residual errors across the past iterations. Intuitively speaking, the accumulated residual errors in both $\lambda_{j|i}^k$ and $\lambda_{j|i}^{k-1}$ would softly constrain $x_i^{k+\frac{1}{2}}$ in a region that incurs small consensus discrepancy with regard to the predefined edge-constraints.

## 4 BELIEF-ATTENTION VIA ORTHOGONAL PROJECTIONS

In this section, we first briefly revisit the standard attention in Transformer. We then motivate and present the orthogonal projections in designing belief-attention. Lastly, we briefly discuss the limitations of belief-attention.

## 4.1 REVISITING ATTENTION IN TRANSFORMER

The original work (Vaswani et al., 2017) proposes the encoder-decoder structure in the transformer for NLP applications. The attention-FFN framework is slightly different in encoder and decoder. For the purposes of demonstration, we consider a simplified version of attention, represented as (see (MHA, 2023; Dosovitskiy et al., 2021))

$$\mathrm{H}_m(X) = \mathrm{attention}(\overbrace{XW_m^Q}^{Q_m}, \overbrace{XW_m^K}^{K_m}, \overbrace{XW_m^V}^{V_m}) \quad m = 1, \ldots, M \tag{6}$$

$$\mathrm{MH}(X) = \mathrm{Concat}(\mathrm{H}_1(X), \ldots, \mathrm{H}_M(X)) \tag{7}$$

$$X \Leftarrow \underbrace{X + \overbrace{\mathrm{MH}(X)W^o}^{\text{linear mapping}}}_{\text{skip-connection}} \tag{8}$$

where the tensor $X \in \mathbb{R}^{n \times d}$ is the input from the layer below in Transformer, $(W_m^Q, W_m^K, W_m^V)$ are the three learnable matrices for computing $(Q_m, K_m, V_m) \in (\mathbb{R}^{n \times d_m}, \mathbb{R}^{n \times d_m}, \mathbb{R}^{n \times d_m})$ of the $m$th attention, and Concat($\ldots$) stacks up $M$ attentions $\{\mathrm{H}_m(X)\}_{m=1}^M$, which is further processed by the linear mapping $W^o$ for post-processing. Lastly, the notations H and MH stand for "head" and "multi-head", respectively.

To facilitate the discussion of attention and QA-PDMM later on, we first briefly explain the notations in (6)-(8). Following the convention of python-based implementation (e.g., pytorch) of attention, the tensor $X \in \mathbb{R}^{n \times d}$ has $n$ tokens and each token is of dimension $d$. We use the row vector $X[i,:]$ to denote the $i$th token. Therefore, the computation for $(Q_m, K_m, V_m)$ in (6) and the linear mapping in (8) are conducted in a token-wise manner.

It is well-known that the attention operation in Equ. (6) is a QK-softmax-based weighted summation of the $n$ row vectors in $V_m$, given by

$$\mathrm{H}_m(X) = \mathrm{softmax} \overbrace{\left( \frac{Q_m K_m^T}{\sqrt{d_m}} \right) V_m}^{\text{info. aggregation}} \tag{9}$$

where $d_m$ is the dimension of the row vectors in $Q_m$. The softmax term computes the unified relevance of each token with respect to neighboring tokens, which generally stabilizes the training process in comparison to other forms of weighted summation. Similarly to that of QA-PDMM, the computed weighted summation of the $n$ row vectors in $V_m$ can be taken as information aggregation from all neighbors.

In general, for a standard non-causal attention, every two tokens are neighbors, which corresponds to a fully connected graph in distributed optimization. On the other hand, a non-casual attention is actually associated with a sparse directed graph. This is because only earlier tokens could make contributions to the current considered token. We will not discuss those different types of graphs in relation with different types of attentions in detail, which is out of the scope in this paper.

## 4.2 UPDATE EXPRESSION OF BELIEF-ATTENTION

**Similarity between QA-PDMM and Transformer layer**: As demonstrated in Fig. 1, both QA-PDMM and the Transformer layer have two skip connections. The variable $x_i$ at node $i$ in QA-PDMM corresponds to the $i$th token $X[i,:]$ in the Transformer layer. The computation for $x_i^{k+\frac{1}{2}}$ in (2) shares a similarity with the computation for $X[i,:]$ in (8), where both quantities aggregate information from neighboring nodes (i.e., other tokens) before performing skip-connection. Finally, from a high-level point of view, $x_i^{k+1}$ in (3) and the final output $X[i,:]$ (see Fig. 1) from a Transformer layer can be viewed as performing local information fusion and skip-connection.

**Motivation**: As discussed earlier, $x_i^{k+\frac{1}{2}}$ in QA-PDMM is updated by exploiting the consensus discrepancy in the form of residual errors of the linear equality constraints in its update expression, which are captured by $\{\lambda_{j|i}^k, \lambda_{j|i}^{k-1}\}$. However, in the expression (8) for the standard attention, the tensor $\mathrm{MH}(X)$ is not really a residual signal from the perspective of distributed optimization. This is because each term $\mathrm{H}_m(X)$ within $\mathrm{MH}(X)$ does not actually measure any discrepancy among the tokens. We argue that the Transformer architecture would benefit if certain type of discrepancy

could be captured by the attention layer. By doing so, the learnable parameters in Transformer could promptly respond to the discrepancy among the tokens during the training process, thus making the learning procedure more efficient. We define $V(X)$ to be

$$V(X) = \text{Concat}(V_1, \ldots, V_M).$$

Furthermore, we use $V(X)[i, :]$ and $\text{MH}(X)[i, :]$ to denote the original V vector and the weighted summation of $V$ vectors for the $i$th token. The remaining step is to define a proper residual signal for the $i$th token in terms of $V(X)[i, :]$ and $\text{MH}(X)[i, :]$ in the attention layer.

**Taking perpendicular component after orthogonal projection as residual signal**: In general, the two vectors $V(X)[i, :]$ and $\text{MH}(X)[i, :])\}$ would be correlated. We let the residual signal be in the following format:

$$\Delta(X)[i, :] = \text{MH}(X)[i, :] - \alpha_i V(X)[i, :], \tag{10}$$

where $\alpha_i$ is a scalar parameter to be specified. We let the optimal solution $\alpha_i^*$ be the one that minimizes the squared norm of $\Delta(X)[i, :]$:

$$\alpha_i^* = \arg\min_{\alpha_i} \|(\Delta(X)[i, :])\|^2, \tag{11}$$

where we use $\|\cdot\|$ to denote the $l_2$ norm of a vector. It is immediate that[1]

$$\alpha_i^* = \frac{\langle \text{MH}(X)[i, :], V(X)[i, :] \rangle}{\langle V(X)[i, :], V(X)[i, :] \rangle}. \tag{12}$$

where $\langle \cdot, \cdot \rangle$ denotes the inner product of two vectors.

By plugging the expression (12) into (10), it is not difficult to show that $\Delta(X)[i, :]$ is orthogonal to $V(X)[i, :]$. That is, $\Delta(X)[i, :]$ is the perpendicular component after orthogonal projection of $\text{MH}(X)[i, :]$ onto $V(X)[i, :]$ (see Fig. 2 for demonstration). By using algebra, one can easily show that the magnitude $\|\Delta(X)[i, :]\|$ is either small or equal to $\|\text{MH}(X)[i, :]\|$.

$$\|\Delta(X)[i, :]\| \le \|\text{MH}(X)[i, :]\|. \tag{13}$$

The perpendicular component $\Delta(X)[i, :]$ reflects a belief about the discrepancy between the original vector (alternatively referred to as *prior-representation*) $V(X)[i, :]$ and the newly computed vector $\text{MH}(X)[i, :]$ (alternatively referred to as *post-representation*) which is obtained via softmax-based weighted summation. In other words, the vector $\Delta(X)[i, :]$ measures a certain discrepancy between the prior- and post-representations of the $i$th token. A large magnitude of $\Delta(X)[i, :]$ indicates that the associated token should be adjusted significantly, and vice versa.

Intuitively, $\Delta(X)[i, :]$ in belief-attention corresponds to the residual signal $\sum_{j \in \mathcal{N}_i} A_{ij}^T (\lambda_{j|i}^k - \lambda_{j|i}^{k-1}) - \rho A_{ji}(x_j^k - x_j^{k-1}))\big)$ in (20) for QA-PDMM. We note that no orthogonal projection is performed in QA-PDMM for computing the residual signal. This is because pre-defined linear constraints (see Equ. 1) are introduced in distributed optimization. On the contrary, in Transformer, we need to define a proper residual signal as specified in (10) and (12) since no constraints are introduced beforehand.

We take the residual signal $\Delta(X)$ to replace $\text{MH}(X)$ when performing the skip-connection operation. The token tensor $X$ can thus be updated as follows:

$$X \Leftarrow \underbrace{X + \overbrace{\Delta(X)W^o}^{\text{linear mapping}}}_{\text{skip-connection}}, \tag{14}$$

where the linear mapping $W^o$ is applied to the residual signal $\Delta(X)$ for post-processing. Fig. 3 demonstrates the pytorch code for realizing belief-attention. In practice, one can easily adopt the pytorch code to convert a standard attention into belief-attention. The main difference with respect to the standard attention is to subtract a scaled version of the original V tensor from the multi-head attention as represented in (12).

---

[1]In our implementation for belief-attention and its variant belief-attention$^*$, we found that there is no need to introduce a small positive value $\epsilon$ to the denominator of $\alpha_i$ to avoid division by zero.

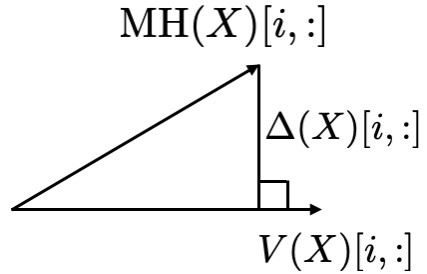

Figure 2: Orthogonal projection to obtain the perpendicular component $\Delta(X)[i,:]$ as residual signal.

```
#linear mapping
Wo = torch.nn.Linear(d_in,d)
##########################
# MH: multi-head attention
# V:  original V tensor
num = torch.sum(MH*V,dim=-1)
den = torch.sum(V*V,dim=-1)
Delta = MH - (num/den)*V
X = X + Wo(Delta)
```

Figure 3: Demonstration of pytorch code for belief-attention.

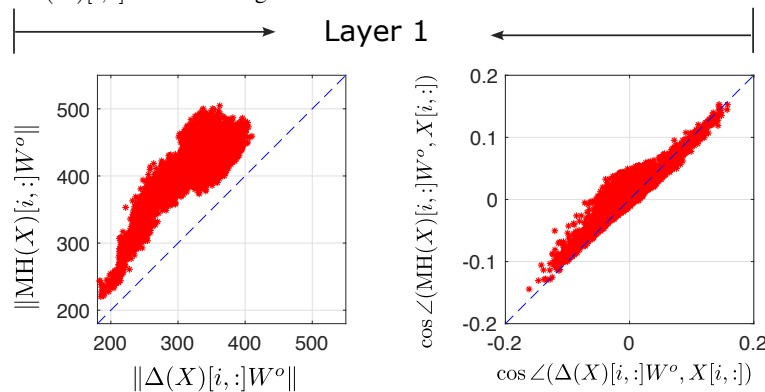

Figure 4: Demonstration of the impact of the perpendicular component $\Delta X[i,:]$. The notation $\angle(\cdot,\cdot)$ stands for the angle between two vectors. The data points in the above four plots are collected when training BeliefFormer of 12 belief-attention layers over ImageNet for the 1st epoch (see Fig. 8 in Appendix A for demonstration of the data points in layer 1 and 10).

We note that layer normalization (LN) or RMSNorm in Transformer directly affect the magnitudes of tokens by standardizing their feature vectors. As demonstrated in Fig. 4, by taking the perpendicular component $\Delta(X)[i,:]$ as the residual signal, the tokens are updated more in their tangent directions and less in their magnitudes. Intuitively, this prevents LN or RMSNorm from diminishing the effects that belief-attention has on token updates.

**Impact of the update expression (14)**: We note that because of the linear mapping $W^o$ in (14) and $\{W_m^V\}_{m=1}^M$ in (6), the transformed residual signal $\Delta(X)W^o$ will not be orthogonal to $X$. That is, the orthogonality would not be not preserved in the token space. We have conducted empirical analysis to investigate the impact of $\Delta(X)W^o$ in comparison to $MH(X)W^o$ in the token space.

The obtained empirical results in Fig. 4 confirm that the magnitudes $\|\Delta(X)[i,:]W^o\|$ are considerably smaller than $\|MH(X)[i,:]W^o\|$. This indicates that the magnitude inequality (13) is preserved by the linear mapping $W^o$. The plots in the figure also demonstrate that the two angles $\angle(\Delta(X)[i,:]W^o, X[i,:])$, and $\angle(MH(X)[i,:]W^o, X[i,:])$ are roughly the same.

The above analysis indicates that the update in (14) with belief-attention indeed causes relatively more change in the tangent directions of the tokens (which are the row vectors in $X$) and less change in their magnitudes, compared to the update in the standard attention.

### 4.3 LIMITATIONS OF BELIEF-ATTENTION W.R.T. ATTENTION

Before we discuss the limitations, we first emphasize that belief-attention does not introduce additional training parameters. That is, both belief-attention and the standard attention have the same number of parameters. Considering the time complexities, as belief-attention requires the additional orthogonal projection operations, it naturally leads to higher training and inference complexities. In the experimental section, we have quantitatively measured the training and/or inference complexities (see Table 1 and 2). It is found that the computational complexities are only slightly increased in comparison to those of the standard attention.

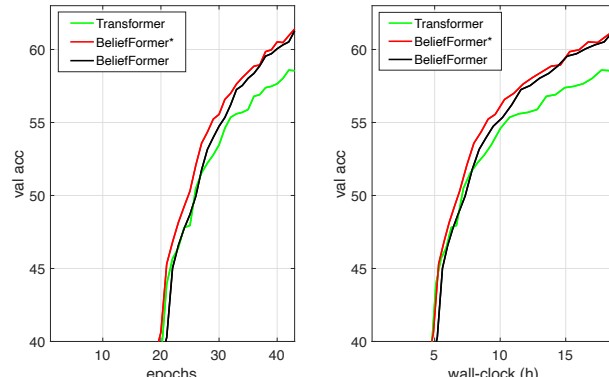

Figure 5: Performance comparison for image classification over ImageNet of 1000 classes.

It is noted that the orthogonal projection is performed on a per-token basis. Therefore, in principle, its computation can be parallelized by using a set of GPUs to reduce the execution time. When performing inference for an auto-regressive BeliefFormer, the time complexity overhead should be linearly proportional to the sequence length.

## 5 A VARIANT OF BELIEF-ATTENTION

In this section, we propose a variant of belief-attention by incorporating both the per attention-based and global orthogonal projections. Firstly, we note that in Subsection 4.2 (see also Fig. 2), we project the entire vector $\text{MH}(X)[i,:]$ w. r. t. $V(X)[i,:]$ in a global manner. Alternatively, we can also project the per attention-head $H_m(X)[i,:]$ w. r. t. the original subvector $V_m[i,:]$. The associated perpendicular component after orthogonal projection can be expressed as

$$\Delta_m^s(X)[i,:] = H_m(X)[i,:] - \beta_{m,i}V_m[i,:] \text{ where } \beta_{m,i} = \frac{\langle \text{H}_m(X)[i,:], V_m[i,:]\rangle}{\langle V_m[i,:], V_m[i,:]\rangle}, \quad (15)$$

for all $m = 1, \ldots, M$, and $i = 1, \ldots, n$. Again, it is found from practice that there is no need to introduce a small positive value to avoid division by zero.

Upon obtaining the two types of perpendicular components in (12) and (15), we then exploit both of them when performing skip-connection. Our main purpose for doing so is to improve the performance of the overall neural architecture with both the global and per attention-head based discrepancies instead of one in belief-attention introduced earlier. The final update expression for $X$ can be represented as

$$\Delta_s(X) = \text{Concat}(\Delta_1^s(X), \ldots, \Delta_M^s(X)) \quad (16)$$

$$X \Leftarrow \underbrace{X + \Delta(X)W^o + \Delta_s(X)W^s}_{\text{skip-connection}}, \quad (17)$$

where $\Delta_s(X)$ is obtained by stacking up $M$ individual residual signals $\{\Delta_m^s(X)\}_{m=1}^M$. In comparison to (14), an additional linear mapping $W^s$ is required to make dimensionality alignment for $\Delta_s(X)$.

In brief, the update expressions (6)-(7), (12), and (15)-(17) together define a new type of attention layer, which we refer to as belief-attention$^*$. Consequently, Transformer equipped with belief-attention$^*$ is referred to as BeliefFormer$^*$. Based on the python code in Fig. 3 for belief-attention, one can easily develop the python code for belief-attention$^*$.

### 5.1 LIMITATIONS OF BELIEF-ATTENTION$^*$

Apparently, belief-attention$^*$ introduces an additional set of learnable parameters in $W^s$ in comparison to the standard attention. Furthermore, since belief-attention$^*$ needs to perform both the per attention-head based and global orthogonal projections, its computational complexity would be slightly higher than that of belief-attention. The results in Table 1 and 2 indicate that the overhead introduced in BeliefFormer$^*$ is acceptable given the fact that its performance gain w. r. t. that of Transformer (see Fig. 5 and 6) is remarkable.

|  | no. of parameters | time for evaluating val. dataset (s) |
|---|---|---|
| Transformer | 22.2M (-) | 37.46 (-) |
| BeliefFormer | 22.2M (0%) | 37.93 (1.3%) |
| BeliefFormer* | 24.0M (8.1%) | 38.87 (3.8%) |

Table 1: Comparison of number of training parameters and computational complexities for image classification over ImageNet. We note that the input image size to the models changes dynamically over training time. Therefore, it is not feasible to measure the average training time per epoch. The values in the round bracket $(\cdot)$ account for the overhead of BeliefFormer* and BeliefFormer in comparison to Transformer in percentage.

|  | no. of parameters | training time (s) per iteration | tokens/s in training | inference time (s) per iteration |
|---|---|---|---|---|
| Transformer | 123.5M (–) | 0.274 (–) | 2284 | 0.237 (–) |
| BeliefFormer | 123.5M (0%) | 0.284 (3.6%) | 2245 | 0.241 (1.7%) |
| BeliefFormer* | 130.6M (5.7%) | 0.299 (9.1% ) | 2147 | 0.260 (9.7%) |

Table 2: Comparison of number of parameters and computational complexities for NLP. Transformer in the table is in fact nano-GPT2. The values in the round bracket $(\cdot)$ account for the overhead of BeliefFormer* and BeliefFormer in comparison to Transformer in percentage.

# 6  EXPERIMENTS

We evaluated BeliefFormer and its variant BeliefFormer* for three tasks: (1) image classification over ImageNet; (2) NLP for training nano-GPT2; (3) NLP for training Llama; (4) image classification over CIFAR10 and CIFAR100. Our experiments make use of four open-source repositories for the above three tasks, which are listed in Table 4 in the appendix. All the experiments were conducted on a computer with a single Nvidia Geforce A6000 GPU with 48GB memory.

In brief, it is found that both BeliefFormer and BeliefFormer* outperform Transformer consistently in first two tasks. BeliefFormer is not tested for the 3rd and 4th task due to limited time in the rebuttal period. BeliefFormer* needs to introduce a small percentage of learnable parameters and marginal computational complexity in comparison to Transformer. BeliefFormer, on the other hand, only introduces marginal computational complexity.

## 6.1  IMAGE CLASSIFICATION OVER IMAGENET

We adopted the 1st open-source repository in Table 4, which is for training a ViT over ImageNet (from 2012). There are 12 attention layers in the original ViT model (the model name is deit_small_patch16_224). We replaced each standard attention in ViT with belief-attention and belief-attention*, respectively. All the models were trained from scratch by using the ImageNet training data. The training setups in terms of the hyper-parameters follow directly from the original open source. After training, they are evaluated via the associated validation dataset.

Fig. 5 visualizes the obtained validation accuracy curves over epochs and over wall-clocks. It is clear that both BeliefFormer and BeliefFormer* outperforms Transformer (which is in fact the ViT model) significantly as the epoch index increases. The right plot in the figure against wall-clock suggests that the additional training time introduced in the two new models is negligible.

Table 5.1 summarizes the number of parameters and inference time for the three models. It is seen that the inference time for the three models are roughly the same when evaluating the valuation dataset, indicating that the orthogonal projection in the two new models can be efficiently computed by using GPU. For this particular task, BeliefFormer* introduces about 8% new parameters to handle two types of orthogonal projections.

## 6.2  NLP FOR TRAINING NANO-GPT2

We adopted the 2nd open-source repository in Table 4 for this experiment. The open-source is for training nano-GPT2 over 5B tokens extracted from OpenWebText. Similarly, we replaced the

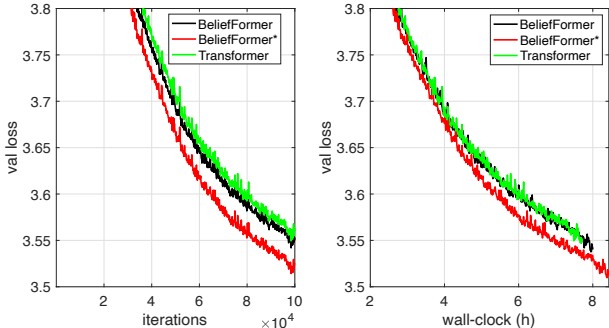

Figure 6: Performance comparison for NLP using 5B tokens extracted from OpenWebText. Transformer in the figure is in fact nano-GPT2.

standard attention layer in nano-GPT2 with belief-attention and belief-attention*, respectively. The training setups follow directly from the original open source. We refer to nano-GPT2 as Transformer in the context below.

Fig. 6 visualizes the validation loss curves over iterations and over wall-clock. Apparently, BeliefFormer* performs significantly better than the other two models even considering wall-clock instead of iterations. This suggests that it is indeed beneficial to include those two types of orthogonal projections as studied in Section 5. On the other hand, BeliefFormer performs slightly better than Transformer across iterations. If the training time complexity is taken into account, BeliefFormer and Transformer have a similar training speed.

Table 2 summarizes the number of parameters and time complexities of the three considered models. Similarly to the 1st task, BeliefFormer* slightly increases the number of parameters and computational complexity, but yields a noticeable improvement in validation performance. Considering BeliefFormer, it introduces only a small overhead in terms of computational complexity.

**Remark 1.** *One may think that the performance gain of belief-attention* over attention in Fig. 6 could be due to the additional linear mapping $W^s$ introduced in (17). To gain deeper insight into belief-attention*, we have also evaluated the performance of another attention layer by concatenating $MH(X)$ and $V$ together, which is then processed by a linear mapping of a larger size as in belief-attention*. It is found that the attention layer by concatenating $MH(X)$ and $V$ produces slightly worse performance than the standard attention at a later training stage. See Appendix C for details.*

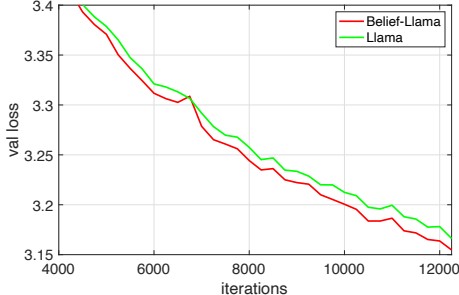

Figure 7: Performance comparison for training Llama and Belief-Llama over FineWeb-Edu of size 10BB. Belief-Llama is obtained by replacing the attention layer in Llama with beleif-attention*.

### 6.3 NLP FOR TRAINING LLAMA

In this experiment, we consider training Llama of size 188MB, which has 12 Transformer layers in total. The fourth open-source repository from Table 4 is exploited. It is noted that Llama and nano-GPT2 are slightly different in their architectures. The dataset being used for training and evaluation is the FineWeb-Edu of size 10BT. We replace all the attention layers in Llama with belief-attention*, which is referred to as Belief-Llama. The two models were trained by using the identical training setups as specified in the open-source.

It is clear from Fig. 7 that Belief-Llama outperforms Llama in terms of the validation loss. The results are consistent with those in Fig. 6 for training nano-GPT2.

## 6.4 IMAGE CLASSIFICATION OVER CIFAR10 AND CIFAR100

In this experiment, we adopted the 3rd open-source repository for training ViT over CIFAR10 and CIFAR100 in Table 4. We replaced the standard attention layer by belief-attention* developed in this paper, which referred to as Belief-ViT.

Aside from the modification to belief-attention*, the training setups follow the original open-source implementation. In brief, each model was trained for 100 epoch by using the AdamW optimizer. Three experimental repetitions (with random seeds in $\{0, 50, 100\}$) were performed per training setup to mitigate the effect of randomness.

Table 3 summarizes the obtained validation accuracy. It is clear that Belief-ViT produces considerably higher validation accuracy than ViT. This indicates that the introduced orthogonal projections is a better choice than the softmax-based weighted summation of V vectors when performing the skip-connection in the attention layer.

Table 3: Validation accuracy for image classification over CIFAR10 and CIFAR100.

|  | ViT | Belief-ViT |
|---|---|---|
| CIFAR10 | 93.62±0.11 | **94.17**±0.13 |
| CIFAR100 | 72.75±0.44 | **74.34**±0.12 |

## 7 CONCLUSIONS

In this work, we have proposed belief-attention and belief-attention* to replace attention in Transformer from a distributed optimization perspective. In particular, we first identify a similarity between the update expressions of QA-PDMM and the attention-FFN framework in Transformer. The softmax-based weighted summation in the standard attention can be viewed as information aggregation from neighboring tokens while the FFN operation can be taken as local information fusion. Inspired by QA-PDMM that exploits the consensus discrepancy in its update expressions, we utilize the discrepancy between the weighted summation of $V$ vectors and their orthogonal projections onto the original $V$ vectors when designing the two new variants of attention layer. As demonstrated in Fig. 4, usage of perpendicular components in belief-attention and belief-attention* would make the tokens be updated relatively more in their tangent directions and less in their magnitudes. Experimental results over three tasks indicate that BeliefFormer (aka Transformer with belief-attention) and BeliefFormer* (aka Transformer with belief-attention*) perform consistently better than Transformer in terms of the validation performance.

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

| ImagNet task | `https://github.com/BorealisAI/efficient-vit-training` |
|---|---|
| NLP-nanoGPT | `https://github.com/KellerJordan/modded-nanogpt/tree/casted` |
| CIFAR10 & CIFAR100 | `https://github.com/aanna0701/SPT_LSA_ViT` |
| NLP-Llama | `https://github.com/hengjiUSTC/learn-llm/tree/main/pretrain` |

Table 4: list of open-source repositories expoited in this paper.

## A  REGARDING GENERATION OF FIGURE 4.

We briefly explain how the data points were collected when generating the four plots of Fig. 4. When we trained BeliefFormer over ImageNet, we computed and collected the four quantities $\|\Delta(X)[i,:]\|$, $\|\text{MH}(X)[i,:]\|$, $\cos\angle(\Delta(X)[i,:]W^o, X[i,:])$, and $\cos\angle(\text{MH}(X)[i,:]W^o, X[i,:])$ for a particular token index $i = 0$ across different belief-attention layers and across different iterations in the first epoch. There are in total 12 belief-attention layers in the tested BeliefFormer. The behaviors of the above four quantities are similar across different layers.

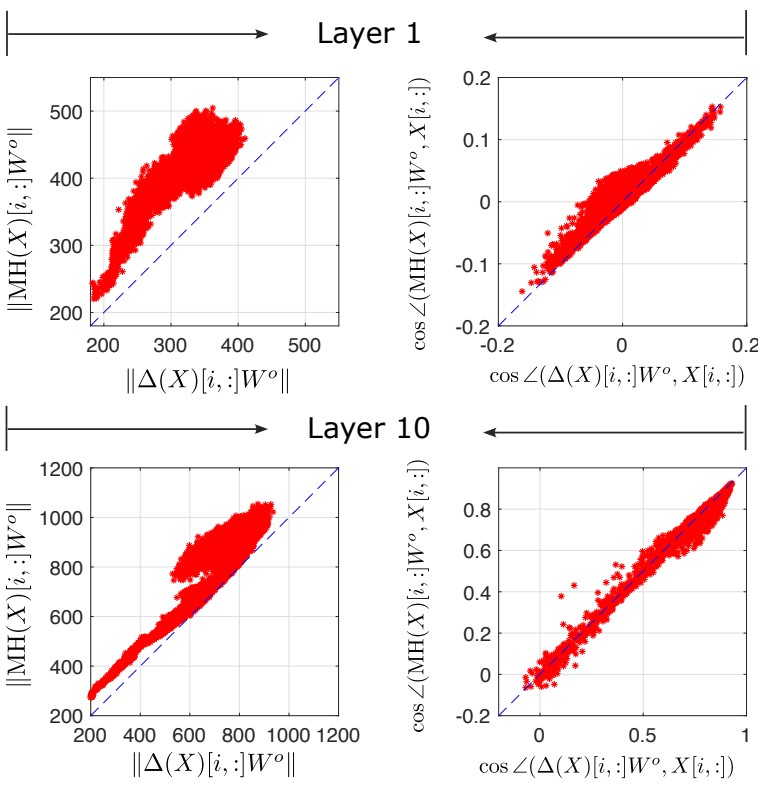

Figure 8: Demonstration of the impact of the perpendicular component $\Delta X[i,:]$. The notation $\angle(\cdot, \cdot)$ stands for the angle between two vectors. The data points in the above four plots are collected when training BeliefFormer of 12 belief-attention layers over ImageNet for the 1st epoch.

## B  DERIVATION OF QA-PDMM

### B.1  UPDATE EXPRESSIONS OF QA-PDMM

In this section, we will explain how to obtain the update expressions (2)-(4) for QA-PDMM starting from the update expressions for PDMM.

PDMM introduces two Lagrangian multipliers $\lambda_{i|j}$ and $\lambda_{j|i}$ for the linear constraint over the edge $(i, j) \in \mathcal{E}$. Let $\mathcal{N}_i$ denote the set of neighbors for node $i$. At the $k$th iteration, each new update $x_i^{k+1}$

is computed in terms of the information $\{(x_{j|i}^k, \lambda_{j|i}^k)|j \in \mathcal{N}_i\}$ from neighbors as

$$x_i^{k+1} = \arg \min_{x_i \in \mathbb{R}^{d_i}} \left[ f_i(x_i) - x_i^T \Big( \sum_{j \in \mathcal{N}_i} A_{ij}^T \lambda_{j|i}^k \Big) + \sum_{j \in \mathcal{N}_i} \frac{\rho}{2} \|A_{ij}x_i + A_{ji}x_j^k - b_{ij}\|^2 \right] \quad \forall i \in \mathcal{V}, \tag{18}$$

where the stepsize $\rho > 0$. Once $x_i^{k+1}$ is available, the associated Lagrangian multipliers of node $i$ are updated to be

$$\lambda_{i|j}^{k+1} = \lambda_{j|i}^k + \rho(b_{ij} - A_{ji}x_j^k - A_{ij}x_i^{k+1}) \quad \forall j \in \mathcal{N}_i. \tag{19}$$

As discussed in the main paper, $\lambda_{i|j}^{k+1}$ is computed by accumulating the residual errors across historical iterates up to iteration $k$ in terms of the linear equality constraint for edge $(i, j) \in \mathcal{E}$.

One potential issue with the update expression (18) is that at iteration $k$, a local optimization problem at each node $i$ needs to be solved to obtain $x_i^{k+1}$. In certain applications (e.g., training a deep neural network (DNN) model), it might be time-consuming or even not possible to obtain an exact solution for $x_i^{k+1}$. In those scenarios, one can perform quadratic approximation based PDMM (QA-PDMM) to simplify the local optimization at each node $i$ and at each iteration $k$.

## B.2  UPDATE EXPRESSIONS OF QA-PDMM

Suppose the gradient $\nabla f_i(x_i)$ for any $x_i \in \mathbb{R}^d$ can be computed in a reasonable amount of time. The basic idea of QA-PDMM is to approximate each local function $f_i(x_i)$ at each iteration $k$ by a quadratic function. The new estimator $x_i^{k+1}$ can then be computed by solving a quadratic optimization problem with $f_i(x_i)$ in (18) being replaced by the quadratic function. We make an assumption below regarding the function $f_i$:

**Assumption 1.** *For each $i \in \mathcal{V}$, assume the gradient $L$-Lipschitz continuous and $m$-strongly convex.*

Next we derive the update expressions (2)-(4) in the main paper for QA-PDMM. We perform the derivation by induction. In particular, we first consider $k = 0$ and $k = 1$ and then extend the derivation to $k \geq 2$.

**Update expression at $k = 0$:** Suppose at iteration $k = 0$, all the primal variables are initialized to be $\{x_i^0\}_{i \in \mathcal{V}}$ and their Lagrangian multipliers are initialized to be $\{\lambda_{i|j}^0|j \in \mathcal{N}_i, i \in \mathcal{V}\}$. We further introduce the initialization of the primal variables $\{x_i^{-\frac{1}{2}} = x_i^0\}_{i \in \mathcal{V}}$ at iteration $k = -\frac{1}{2}$.

We now consider computing $\{x_i^{\frac{1}{2}}\}_{i \in \mathcal{V}}$ at iteration $k = 0$. To do so, we first approximate each $f_i(x_i)$ by a quadratic function in terms of $x_i^{-\frac{1}{2}}$ as

$$f_i(x_i) \approx f_i(x_i^{-\frac{1}{2}}) + \nabla f_i(x_i^{-\frac{1}{2}})^T(x_i - x_i^{-\frac{1}{2}}) + \frac{\eta}{2}\|x_i - x_i^{-\frac{1}{2}}\|^2, \tag{20}$$

where the parameter $\eta$ is chosen to be $\eta \geq L^2/(2m)$. By combining (18) and (20) at iteration $k = 0$, each $x_i^{\frac{1}{2}}$ can be computed by solving the following optimization problem at iteration $k = 0$:

$$x_i^{\frac{1}{2}} = \arg \min_{x_i \in \mathbb{R}^{d_i}} \left[ f_i(x_i^{-\frac{1}{2}}) + \nabla f_i(x_i^{-\frac{1}{2}})^T(x_i - x_i^{-\frac{1}{2}}) + \frac{\eta}{2}\|x_i - x_i^{-\frac{1}{2}}\|^2 \right.$$
$$\left. - x_i^T \left( \sum_{i \in \mathcal{N}_i} A_{ij}^T \lambda_{j|i}^0 \right) + \sum_{j \in \mathcal{N}_i} \frac{\rho}{2} \|A_{ij}x_i + A_{ji}x_j^0 - b_{ij}\|^2 \right] \quad \forall i \in \mathcal{V},$$
$$= \eta B_i^{-1} x_i^{-\frac{1}{2}} + B_i^{-1} \Big( \sum_{i \in \mathcal{N}_i} A_{ij}^T(\lambda_{j|i}^0 - \rho A_{ji}x_j^0 + \rho b_{ij}) - \nabla f_i(x_i^{-\frac{1}{2}}) \Big), \tag{21}$$

where $B_i = (\eta I + \rho \sum_{i \in \mathcal{N}_i} A_{ij}^T A_{ij})$. Apparently, $x_i^{\frac{1}{2}}$ is a function of $x_i^{-\frac{1}{2}}$ and information $\{(x_j^0, \lambda_{j|i}^0)|j \in \mathcal{N}_i\}$ from neighbors.

With $x_i^{\frac{1}{2}}$, we are now ready to compute $x_i^1$ for each node $i \in \mathcal{V}$. In principle, $x_i^{\frac{1}{2}}$ should be a better solution than $x_i^{-\frac{1}{2}}$ for solving the original optimization problem (1). We then approximate each $f_i(x_i)$ by a quadratic function in terms of $x_i^{\frac{1}{2}}$ as

$$f_i(x_i) \approx f_i(x_i^{\frac{1}{2}}) + \nabla f_i(x_i^{\frac{1}{2}})^T(x_i - x_i^{\frac{1}{2}}) + \frac{\eta}{2}\|x_i - x_i^{\frac{1}{2}}\|^2. \tag{22}$$

By combining (18) and (22) at iteration $k = 0$, each $x_i^{\frac{1}{2}}$ can be computed by solving the following optimization problem at iteration $k = 0$:

$$x_i^1 = \arg \min_{x_i \in \mathbb{R}^{d_i}} \left[ f_i(x_i^{\frac{1}{2}}) + \nabla f_i(x_i^{\frac{1}{2}})^T(x_i - x_i^{\frac{1}{2}}) + \frac{\eta}{2}\|x_i - x_i^{\frac{1}{2}}\|^2 \right.$$

$$\left. - x_i^T\left(\sum_{i \in \mathcal{N}_i} A_{ij}^T \lambda_{j|i}^0\right) + \sum_{j \in \mathcal{N}_i} \frac{\rho}{2}\|A_{ij}x_i + A_{ji}x_j^0 - b_{ij}\|^2 \right] \quad \forall i \in \mathcal{V},$$

$$= \eta B_i^{-1}x_i^{\frac{1}{2}} + B_i^{-1}\left(\sum_{i \in \mathcal{N}_i} A_{ij}^T(\lambda_{j|i}^0 - \rho A_{ji}x_j^0 + \rho b_{ij}) - \nabla f_i(x_i^{\frac{1}{2}})\right). \tag{23}$$

The expression (23) for $x_i^1$ is slightly different from (21) for $x_i^{\frac{1}{2}}$. Specifically, $x_i^1$ is a function of $x_i^{\frac{1}{2}}$ and information $\{(x_j^0, \lambda_{j|i}^0)|j \in \mathcal{N}_i\}$ from neighbors.

With $\{x_i^1\}_{i \in \mathcal{V}}$, the new estimation for their associated Lagrangian multipliers can be computed by following (19) as

$$\lambda_{i|j}^1 = \lambda_{j|i}^0 + \rho(b_{ij} - A_{ji}x_j^0 - A_{ij}x_i^1) \quad \forall j \in \mathcal{N}_i, i \in \mathcal{V}.$$

**Update expression at $k = 1$:** We will show that the expressions for $x_i^{\frac{3}{2}}$ and $x_i^2$ coincide with (2)-(3) by specifying $k = 1$. Similarly to iteration $k = 0$, we first compute $x_i^{\frac{3}{2}}$ as a function of $x_i^{\frac{1}{2}}$ and information $\{(x_j^1, \lambda_{j|i}^1)|j \in \mathcal{N}_i\}$ from neighbors:

$$x_i^{\frac{3}{2}} = \arg \min_{x_i \in \mathbb{R}^{d_i}} \left[ f_i(x_i^{\frac{1}{2}}) + \nabla f_i(x_i^{\frac{1}{2}})^T(x_i - x_i^{\frac{1}{2}}) + \frac{\eta}{2}\|x_i - x_i^{\frac{1}{2}}\|^2 \right.$$

$$\left. - x_i^T\left(\sum_{i \in \mathcal{N}_i} A_{ij}^T \lambda_{j|i}^1\right) + \sum_{j \in \mathcal{N}_i} \frac{\rho}{2}\|A_{ij}x_i + A_{ji}x_j^1 - b_{ij}\|^2 \right] \quad \forall i \in \mathcal{V},$$

$$= \eta B_i^{-1}x_i^{\frac{1}{2}} + B_i^{-1}\left(\sum_{i \in \mathcal{N}_i} A_{ij}^T(\lambda_{j|i}^1 - \rho A_{ji}x_j^1 + \rho b_{ij}) - \nabla f_i(x_i^{\frac{1}{2}})\right). \tag{24}$$

Note that the gradient $\nabla f_i(x_i^{\frac{1}{2}})$ appears in both (23) and (24). It is not difficult to show that $x_i^{\frac{3}{2}}$ can be represented in terms of $x_i^1$ and the information $\{(\lambda_{j|i}^1 - \lambda_{j|i}^0)|j \in \mathcal{N}_j\}$ and $\{(x_j^1 - x_j^0)|j \in \mathcal{N}_j\}$ from neighbors as

$$x_i^{\frac{3}{2}} = x_i^1 + B_i^{-1}\sum_{i \in \mathcal{N}_i} A_{ij}^T((\lambda_{j|i}^1 - \lambda_{j|i}^0) - \rho A_{ji}(x_j^1 - x_j^0)). \tag{25}$$

We have proved that (25) for $x_i^{\frac{3}{2}}$ indeed coincides with (2) by letting $k = 1$.

The next step is to derive an expression for $x_i^2$ and then show that it coincides with (3) by letting $k = 1$. Similarly to iteration $k = 0$, $x_i^2$ can be computed by solving the following optimization problem at iteration $k = 1$:

$$x_i^2 = \arg \min_{x_i \in \mathbb{R}^{d_i}} \left[ f_i(x_i^{\frac{3}{2}}) + \nabla f_i(x_i^{\frac{3}{2}})^T(x_i - x_i^{\frac{3}{2}}) + \frac{\eta}{2}\|x_i - x_i^{\frac{3}{2}}\|^2 \right.$$

$$\left. - x_i^T\left(\sum_{i \in \mathcal{N}_i} A_{ij}^T \lambda_{j|i}^1\right) + \sum_{j \in \mathcal{N}_i} \frac{\rho}{2}\|A_{ij}x_i + A_{ji}x_j^1 - b_{ij}\|^2 \right] \quad \forall i \in \mathcal{V},$$

$$= \eta B_i^{-1}x_i^{\frac{3}{2}} + B_i^{-1}\left(\sum_{i \in \mathcal{N}_i} A_{ij}^T(\lambda_{j|i}^1 - \rho A_{ji}x_j^1 + \rho b_{ij}) - \nabla f_i(x_i^{\frac{3}{2}})\right). \tag{26}$$

Again note that (24) and (26) share a common quantity $B_i^{-1}\Big(\sum_{i\in\mathcal{N}_i} A_{ij}^T(\lambda_{j|i}^1 - \rho A_{ji}x_j^1 + \rho b_{ij}))\Big)$. Therefore, $x_i^2$ can be represented in terms of $x_i^{\frac{3}{2}}$ and $x_i^{\frac{1}{2}}$ as

$$x_i^2 = x_i^{\frac{3}{2}} + B_i^{-1}(\eta(x_i^{\frac{3}{2}} - x_i^{\frac{1}{2}}) - (\nabla f_i(x_i^{\frac{3}{2}}) - \nabla f_i(x_i^{\frac{1}{2}}))). \tag{27}$$

It is immediate that the expression (27) coincides with (3) by letting $k = 1$.

With $\{x_i^2\}_{i\in\mathcal{V}}$, the new estimation for their associated Lagrangian multipliers can be computed by following (19) as

$$\lambda_{i|j}^2 = \lambda_{j|i}^1 + \rho(b_{ij} - A_{ji}x_j^1 - A_{ij}x_i^2) \quad \forall j \in \mathcal{N}_i, i \in \mathcal{V}.$$

**Update expression at iteration** $k$: Suppose at iteration $k \geq 2$, we have already obtained $\{(x_i^k, x_i^{k-\frac{1}{2}})|i \in \mathcal{V}\}$ and $\{\lambda_{j|i}^k|j \in \mathcal{E}, i \in \mathcal{V}\}$ with

$$x_i^{k-1+\frac{1}{2}} = \eta B_i^{-1} x_i^{k-2+\frac{1}{2}} + B_i^{-1}\Big(\sum_{i\in\mathcal{N}_i} A_{ij}^T(\lambda_{j|i}^{k-1} - \rho A_{ji}x_j^{k-1} + \rho b_{ij}) - \nabla f_i(x_i^{k-2+\frac{1}{2}})\Big) \tag{28}$$

$$x_i^k = \eta B_i^{-1} x_i^{k-1+\frac{1}{2}} + B_i^{-1}\Big(\sum_{i\in\mathcal{N}_i} A_{ij}^T(\lambda_{j|i}^{k-1} - \rho A_{ji}x_j^{k-1} + \rho b_{ij}) - \nabla f_i(x_i^{k-1+\frac{1}{2}})\Big), \tag{29}$$

for all $i \in \mathcal{V}$.

We need to derive the update expression for next iteration. Firstly, the estimates $\{x_i^{k+\frac{1}{2}}|k \in \mathcal{V}\}$ can be computed to be

$$x_i^{k+\frac{1}{2}} = \arg\min_{x_i \in \mathbb{R}^{d_i}} \Big[\nabla f_i(x_i^{k-\frac{1}{2}})^T(x_i - x_i^{k-\frac{1}{2}}) + \frac{\eta}{2}\|x_i - x_i^{k-\frac{1}{2}}\|^2$$

$$- x_i^T(\sum_{i\in\mathcal{N}_i} A_{ij}^T\lambda_{j|i}^k) + \sum_{j\in\mathcal{N}_i} \frac{\rho}{2}\|A_{ij}x_i + A_{ji}x_j^k - b_{ij}\|^2\Big]$$

$$= \eta B_i^{-1} x_i^{k-\frac{1}{2}} + B_i^{-1}\Big(\sum_{i\in\mathcal{N}_i} A_{ij}^T(\lambda_{j|i}^k - \rho A_{ji}x_j^k + \rho b_{ij}) - \nabla f_i(x_i^{k-\frac{1}{2}})\Big) \tag{30}$$

$$\overset{(a)}{=} x_i^k + B_i^{-1}\Big(\sum_{i\in\mathcal{N}_i} A_{ij}^T(\lambda_{j|i}^k - \lambda_{j|i}^{k-1}) - \rho A_{ji}(x_j^k - x_j^{k-1}))\Big), \tag{31}$$

where step $(a)$ is derived by utilizing (29).

Upon obtaining $\{x_i^{k+\frac{1}{2}}|k \in \mathcal{V}\}$, the expressions for $\{x_i^{k+1}|k \in \mathcal{V}\}$ can be derived as

$$x_i^{k+1} = \arg\min_{x_i \in \mathbb{R}^{d_i}} \Big[\nabla f_i(x_i^{k+\frac{1}{2}})^T(x_i - x_i^{k+\frac{1}{2}}) + \frac{\eta}{2}\|x_i - x_i^{k+\frac{1}{2}}\|^2$$

$$- x_i^T(\sum_{i\in\mathcal{N}_i} A_{ij}^T\lambda_{j|i}^k) + \sum_{j\in\mathcal{N}_i} \frac{\rho}{2}\|A_{ij}x_i + A_{ji}x_j^k - b_{ij}\|^2\Big]$$

$$= \eta B_i^{-1} x_i^{k+\frac{1}{2}} + B_i^{-1}\Big(\sum_{i\in\mathcal{N}_i} A_{ij}^T(\lambda_{j|i}^k - \rho A_{ji}x_j^k + \rho b_{ij}) - \nabla f_i(x_i^{k+\frac{1}{2}})\Big)$$

$$\overset{(a)}{=} x_i^{k+\frac{1}{2}} + B_i^{-1}(\eta(x_i^{k+\frac{1}{2}} - x_i^{k-\frac{1}{2}}) - (\nabla f_i(x_i^{k+\frac{1}{2}} - \nabla f_i(x_i^{k-\frac{1}{2}})), \tag{32}$$

where step $(a)$ follows from (30).

It is immediate that (31)-(32) coincide with (2)-(3). The update expression (4) for the Lagrangian multipliers follows directly from (19) for PDMM. The proof is complete.

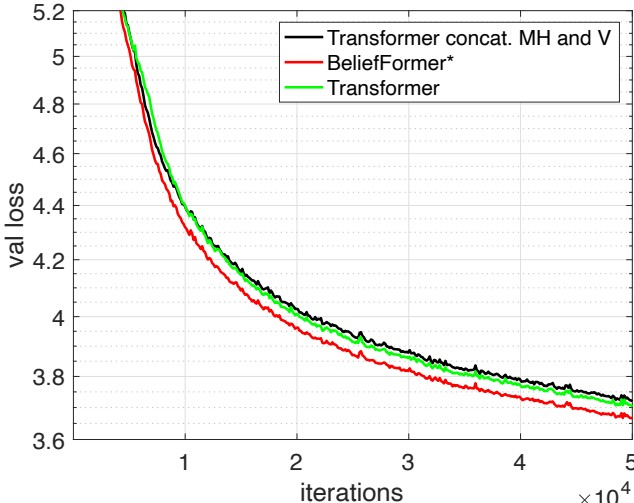

Figure 9: Performance comparison for NLP using 5B tokens extracted from OpenWebText. Transformer in the figure is in fact nano-GPT2.

## C  PERFORMANCE OF AN ATTENTION LAYER BY CONCATENATING $\text{MH}(X)$ AND $V(X)$

In this section, we present additional experimental results for training a Transformer with another attention layer obtained by concatenating both $\text{MH}(X)$ and $V(X)$, which is processed by a linear mapping of a larger size than that the one in the standard attention layer. In particular, we considered the NLP task as presented in the main paper. In this case, the number of parameters in belief-attention* is the same as in the new attention layer by concatenating both $\text{MH}(X)$ and $V(X)$.

Fig. 9 visualizes the validation loss curves over iterations for three different attention layers. The loss curves for Transformer and BeliefFormer* were obtained from Fig. 6 directly. In brief, it is found in the beginning of the training procedure, the attention layer by concatenating both $\text{MH}(X)$ and $V(X)$ produces lower validation performance (i.e., the black curve in Fig. 9) than the standard attention. However, as the iteration increases, the performance gain keeps decreasing. Starting from 2k iterations, the new attention layer performs slightly worse than the standard attention. This could be explained by the fact that the $V$ tensor does not really provide new information from other tokens. When we concatenate $\text{MH}(X)$ and $V(X)$ for linear mapping and skip-connection in each attention layer, it may confuse the overall Transformer architecture.

The above experiment indicates that the orthogonal projections introduced in BeliefFormer* make a difference for performance improvement.

