# OpenReview forum: "BeliefFormer: Belief Attention in Transformer"
_ICLR.cc/2026/Conference — ICLR 2026 Conference Withdrawn Submission_

### Official Review · Reviewer_j7jm · 2025-10-26

**Soundness:** 1
**Presentation:** 3
**Contribution:** 1
**Rating:** 2
**Confidence:** 4

**Summary:**

The paper replaces the usual attention residual MH(X)Wo with an orthogonalized residual that projects the softmax-weighted value summation onto the subspace orthogonal to the original V’s direction, argued to update tokens tangentially and shows modest gains on ImageNet, OpenWebText, and CIFAR-10. A second variant (Belief-Attention*) further orthogonalizes each head’s output w.r.t. its own
Vm, concatenates these per-head tensors and applies an extra projection Ws, yielding slightly larger gains at a small cost in parameters and compute.

**Strengths:**

1) Both the introduced Variants are easy to implement in PyTorch/Jax.
2) The new variants don't introduce heavy compute/wall-clock overhead.

**Weaknesses:**

The paper lacks empirical validation; values are reported at early to mid-training checkpoints, and the models do not appear to be well-tuned. Typically, well-tuned 20M ViT scores at least 70% accuracy on ImageNet classification, but the paper reports numbers close to 60%. It is the same with CIFAR-10, we expect the model to score above 90-95% but the paper's numbers are below 90. In addition, the paper didn't evaluate Language models' performance on downstream tasks. The paper also produced no empirical evidence that the proposed variants scale.

**Questions:**

How can we conclude that the performance improvements of Belief-attention* come from orthogonalization rather than simply increasing the number of parameters? For example, in some of my experiments, concatenating (MH, V) along dim = −1 and feeding it to Wo improves performance. In this setup, the input dimension of  Wo doubles, matching the parameter count of Belief-attention*. Providing ablations of parameters-matched variants like the above would increase confidence that orthogonalization is the source of the improvement.

---

> ### Author Response · Authors · 2025-11-25
>
> __* How can we conclude that the performance improvements of Belief-attention* come from orthogonalization rather than simply increasing the number of parameters?  For example, in some of my experiments, concatenating (MH, V) along dim = -1 and feeding it to Wo improves performance.__
>
> Many thanks for the insightful comments. By following your suggestions, we performed an additional experiment for training nano-GPT by concatenating (MH, V) along dim = -1 in attention, which is then linearly mapped by a $W_o$ of a larger size.  We found that in the beginning of the training procedure, the validation performance is indeed better than the standard Transformer. However, as the iteration increases, the performance gain keeps decreasing. After a certain number of iterations (about 5k), the validation performance becomes worse than the standard Transformer. This could be explained by the fact that the V tensor does not really provide new information from neighbors. We will include the results in the revision later on.
>
> The above experiments indicate that it is the orthogonalisation introduced in belief-attention$^{\ast}$ that makes a difference to obtain better validation performance.
>
>
> __* Regarding the experiments on ImageNet and Cifar10.__
>
> Many thanks for the comments. We really appreciate if the reviewer could provide the links to the related open-sources for ViT over ImageNet and Cifar10 so we can conduct additional experiments to verify the effectiveness of belief-attention and belief-attention$^{\ast}$.
>
> For the open-sources we found and tested in the paper, the performance of the standard Transformer in our experiments coincide with the reported results in those open-source repositories. The experimental comparison shows that belief-attention and belief-attention$^{\ast}$ leads to consistent performance gains.
>
>
> __*  Regarding the large scale experiments.__
>
>
> Many thanks for the comments. Due to limited GPU resources and limited time at the moment, we were not able to conduct experiments for larger-scale language models.
>
> On the other hand, we fully agree with the reviewers that it is of high interest to find out if the reported gains also hold for larger-scale language models or not. We will perform the large-scale experiment (e.g., training lage scale llama models) in  the next step by renting the GPU resources from either Google colab or from other platforms.

---

> > ### Comment · Reviewer_j7jm · 2025-11-26
> >
> > Thanks for the comments. I have a couple of follow-up questions.
> >
> > 1. I wasn’t able to find the above-mentioned experiment in the manuscript. Could you clarify whether it has been added yet?
> >
> > 2. For ImageNet‑1k, a useful reference for typical performance of plain ViT baselines is **“Better plain ViT baselines for ImageNet‑1k”** by Beyer, Zhai, and Kolesnikov [1]. Check their code in the `big_vision` repository
> >  (https://github.com/google-research/big_vision) that reach around 76–80% top‑1 on ImageNet‑1k with ViT-S/16. Using a similar training setup or discussing why your results differ from these standard baselines would strengthen the paper's claims.
> >
> > [1] Lucas Beyer, Xiaohua Zhai, Alexander Kolesnikov. *Better plain ViT baselines for ImageNet‑1k.* https://arxiv.org/abs/2205.01580

---

> > > ### Author Response · Authors · 2025-11-26
> > >
> > > Thank you very much for checking our responses and providing feedbacks in such a short time.
> > >
> > > 1. We have updated the revision to include the additional experiments for testing an attention layer by concatenating both MH and V.  See Remark 1 on page 10 and Appendix C.  You are welcome to provide more suggestions for testing the effectiveness of belief-attention$^{\ast}$.
> > >
> > > 2. Many thanks for the providing the open-source links. We will see if we are able to use a single A6000 GPU  to start training ViT-S/16.
> > >
> > > 3. If you also know good open-source repositories regarding training ViT over CIFAR10, that would be great. We can then easily use a single A6000 GPU to perform this small scale experiment.

---

> > > > ### Comment · Reviewer_j7jm · 2025-11-26
> > > >
> > > > Thank you for updating the manuscript.
> > > >
> > > > For CIFAR-10 and small-sized datasets, I would suggest the "Vision Transformer for Small-Size Datasets" [2] - repository (https://github.com/aanna0701/SPT_LSA_ViT). They report 93% accuracy on CIFAR-10 with ViT.
> > > >
> > > > [2] Seung Hoon Lee, Seunghyun Lee, Byung Cheol Song. Vision Transformer for Small-Size Datasets.
> > > > https://arxiv.org/abs/2112.13492

---

> ### Author Response · Authors · 2025-12-03
>
> We have performed additional experiments by using the recommended open-source (https://github.com/aanna0701/SPT_LSA_ViT) for CIFAR10 and CIAR100. It is found again that by replacing the standard attention layer with belief-attention$^{\ast}$ in ViT, the validation accuracy is improved considerably (see the revised manuscript.).  For example, the validation accuracy of ViT and Belief-ViT (our new approach) over CIFAR100 is 72.75\% and 74.34\%, respectively.   The validation accuracy of ViT and Belief-ViT over CIFAR10 is 93.62\% and 94.17\%, respectively.
>
> We have also trained LLama (roughly 181M parameters) and Belief-Llama (again obtained by replacing the standard attention layer with  belief-attention$^{\ast}$) over FineWeb-Edu of size 10BT. The open-source ( https://github.com/hengjiUSTC/learn-llm/tree/main/pretrain) is used for this task. It is found that Belief-Llama produces better validation loss than LLama (see the revised manuscript).
>
> We are now in the process of testing LLama (roughly 1B parameters) and Belief-Llama over FineWeb-Edu of size 10BT by using a single A6000 GPU. So far we have trained the models for 2250 iterations. Again we start to observe performance gain of Belief-Llama over Llama.

---

### Official Review · Reviewer_5MXh · 2025-11-01

**Soundness:** 3
**Presentation:** 3
**Contribution:** 3
**Rating:** 4
**Confidence:** 2

**Summary:**

This paper proposes BeliefFormer, a drop-in modification to the Transformer attention layer that replaces the usual residual signal (the softmax-weighted sum over V) with an orthogonal-projection-based discrepancy between the aggregated value MH(X) and the original value vectors V. Concretely, it computes, per token, the component of MH(X) that is orthogonal to V and uses that as the residual before the output linear map. A variant, BeliefFormer*, adds per-head orthogonal projections and an extra linear map to capture both global and per-head discrepancies. The method is motivated by an analogy to distributed optimization (PDMM), where updates explicitly incorporate constraint residuals; here, the “belief” is the discrepancy between aggregated and original values. The authors argue this leads to updates that change token directions more than magnitudes, potentially improving generalization.

**Strengths:**

1. The PDMM analogy is thoughtfully developed, and the geometric analysis of the orthogonal projection (directional vs magnitude changes) is sound and intuitive, with supportive empirical diagnostics.
2. The change is minimal (a per-token orthogonal projection), adds no parameters for BeliefFormer and only one extra linear map for BeliefFormer*, and can be implemented with a few lines of code as a drop-in replacement.
3. The paper is clearly written and well structured.

**Weaknesses:**

1. The evaluation is limited to small-to-mid-scale settings (ViT-small/DeiT-small, nano-GPT2). There are no results on large-scale LLMs or larger ViT backbones/long-context settings. As a result, the main claim of broadly improved generalization and scalability across Transformers is not convincingly supported.
2. The PDMM analogy remains heuristic, there is no formal mapping of attention+FFN updates to a constrained optimization scheme, no convergence or generalization guarantees, and no theory showing that orthogonal projection necessarily improves optimization stability or generalization. Moreover, the desired “orthogonality” is not preserved after the output linear map W_o, undercutting the claimed mechanism without a formal treatment of how W_o and W_V interact with the geometry.

**Questions:**

See Weakness

---

> ### Author Response · Authors · 2025-11-25
>
> __* The evaluation is limited to small-to-mid-scale settings (ViT-small/DeiT-small, nano-GPT2). There are no results on large-scale LLMs or larger ViT backbones/long-context settings. As a result, the main claim of broadly improved generalization and scalability across Transformers is not convincingly supported.__
>
> Many thanks for the comments. Due to limited GPU resources and limited time at the moment, we were not able to conduct experiments for larger-scale language models.
>
> On the other hand, we fully agree with the reviewers that it is of high interest to find out if the reported gains also hold for larger-scale language models or not. We will perform the large-scale experiment (e.g., training lage scale llama models) in  the next step by renting the GPU resources from either Google colab or from other platforms.
>
> __* Regarding the mapping between  PDMM and a transformer layer.__
>
>
> Many thanks for the comments. We have revised the paper based on the comments. In the revision, we introduce quadratic-approximation based PDMM (QA-PDMM), which share a great structural similarity with a standard Transformer layer. Figure 1 in the revision visualizes the structure similarity between QA-PDMM and the Transformer layer. A new paragraph on page 4 explicitly explains the structural similarity in text.
>
> Basically, at each iteration $k$ in QA-PDMM, the estimate $x_i^{k+\frac{1}{2}}$ at node $i$ is computed by performing information aggregation from neighbors, followed by a skip-connection. $x_i^{k+1}$ is then computed at node $i$ with the local information $x_i^{k+\frac{1}{2}}$ and $x_i^{k-\frac{1}{2}}$, followed by another skip-connection. See the appendix of the revision for derivation of QA-PDMM from PDMM. From a high level point of view, the computation for $x_i^{k+\frac{1}{2}}$ in QA-PDMM corresponds to the attention layer for updating the $i$th token in Transformer and the computation for  $x_i^{k+1}$ corresponds to the FFN layer for updating the $i$th token again.
>
> __*  Moreover, the desired “orthogonality” is not preserved after the output linear map $W_o$, undercutting the claimed mechanism without a formal treatment of how $W_o$ and $W_V$ interact with the geometry.__
>
> Many thanks for the comments.  We have performed empirical study (see Fig.~4 in the revision) to show that when using the perpendicular component as residual signal in belief-attention, the tokens are indeed updated more in their tangent directions and less in their magnitudes.
>
> We also also added a paragraph in the bottom of page 6 to discuss why belief-attentions leads to better generalization performance. In brief, we note that layer normalization (LN) or RMSNorm in Transformer directly affect the magnitudes of tokens by standardizing their feature vectors. As demonstrated in Fig.~4 in the revision, by taking the perpendicular component $\Delta(X)[i,:]$ as the residual signal, the tokens are updated more in their tangent directions and less in their magnitudes. Intuitively, this prevents LN or RMSNorm from diminishing the effects that belief-attention has on token updates.

---

### Official Review · Reviewer_axrZ · 2025-11-03

**Soundness:** 2
**Presentation:** 2
**Contribution:** 2
**Rating:** 4
**Confidence:** 3

**Summary:**

This paper proposes the BeliefFormer architecture (with the base version belief-attention and variant belief-attention∗ ) for the improvement of the attention layer of Transformer. The core idea is to borrow the PDMM algorithm in distributed optimization, and use the orthogonal projection of the weighted sum of the V-vectors and the original V-vectors as the residual signal (instead of the direct weighted sum of the standard attention), so as to make the token vectors update more along the tangent direction and reduce the amplitude change, thus improving the generalization performance. Experiments are verified in ImageNet/CIFAR10 image classification, and nano-GPT2 for NLP tasks, and both variants outperform the standard Transformer, and the base version of BeliefFormer does not add extra parameters.

**Strengths:**

- In 3 types of tasks (image classification, NLP), the verification accuracy/loss of BeliefFormer and variants are better than that of the standard Transformer, with no task adaptation failure problem.
- The scheme is easy to implement and improves performance without introducing additional significant computational overhead increases

**Weaknesses:**

- Slight increase in computational complexity: training/reasoning time for the variants is  higher than the standard Transformer and overhead may accumulate in long sequence scenarios.
- Tested only in ViT (small models), nano-GPT2 and 3 types of basic tasks, generalization to complex tasks such as large LLMs, very long sequences or speech/image generation was not verified. Insufficient validation of applicability.
- The core of the paper I think lies in treating attn as a distribution optimization problem on a connected graph, and therefore (following the logic of the paper) how the difference metric is constructed is key. Then why orthogonal projection is used, the paper does not give a detailed argument for this, which is a problem.
- In addition, the paper tries to explain the attn process by adopting the idea of PDMM for distribution optimization, but this explanation part lacks detailed arguments, and the authors directly regard the residual part of the attn computation as the residual part of the Lagrange multiplier, and again regard the MHA part as the information cohabitation part. So, the question here is, for the input X in attn is it regarded as the optimization objective X. Then is the V-vector regarded as the multiplier? The paper does not describe these details clearly, and the proposed role of the PDMM is confusing, as I could not make a proper connection between the PDMM and the computation of attn in the way it is described in the paper.

**Questions:**

see Weakness

---

> ### Author Response · Authors · 2025-11-25
>
> __* Regarding the connection between PDMM and a standard transformer layer__
>
> Many thanks for the comments. We have revised the paper based on the comments. In the revision, we introduce quadratic-approximation based PDMM (QA-PDMM), which share a great structural similarity with a standard Transformer layer. Figure 1 in the revision visualizes the structure similarity between QA-PDMM and the Transformer layer. A new paragraph on page 4 explicitly explains the structural similarity in text.
>
> Basically, at each iteration $k$ in QA-PDMM, the estimate $x_i^{k+\frac{1}{2}}$ at node $i$ is computed by performing information aggregation from neighbors, followed by a skip-connection. $x_i^{k+1}$ is then computed at node $i$ with the local information $x_i^{k+\frac{1}{2}}$ and $x_i^{k-\frac{1}{2}}$, followed by another skip-connection. See the appendix of the revision for derivation of QA-PDMM from PDMM. From a high level point of view, the computation for $x_i^{k+\frac{1}{2}}$ in QA-PDMM corresponds to the attention layer for updating the $i$th token in Transformer and the computation for  $x_i^{k+1}$ corresponds to the FFN layer for updating the $i$th token again.
>
> __* Slight increase in computational complexity: training/reasoning time for the variants is higher than the standard Transformer and overhead may accumulate in long sequence scenarios.__
>
> We agree that the computational complexity of belief-attention and its variant would indeed be slightly increased, which we have discussed in the paper.
>
>
> On the other hand, the proposed belief-attention and its variant also bring noticeable performance gains as presented in the experimental section. We are not aware of other existing methods that bring noticeable performance gains with only minor modification to the attention layer as we do. From a scientific point of view, our work could shed light on the  design of new and effective variants of attention layer in the future.
>
> __*  Tested only in ViT (small models), nano-GPT2 and 3 types of basic tasks, generalization to complex tasks such as large LLMs, very long sequences or speech/image generation was not verified. Insufficient validation of applicability.__
>
>
> Many thanks for the comments. Due to limited GPU resources and limited time at the moment, we were not able to conduct experiments for larger-scale language models.
>
> On the other hand, we fully agree with the reviewers that it is of high interest to find out if the reported gains also hold for larger-scale language models or not. We will perform the large-scale experiment (e.g., training llama models) in  the next step by renting the GPU resources from either Google colab or from other platforms.
>
> __* Regarding why orthogonal projection is employed in design of belief-attention.__
>
>
> Many thanks for the comments. We have added one more paragraph at the bottom of page 6 to provide an explanation why belief-attention leads to performance improvement.
>
> In brief, we note that layer normalization (LN) or RMSNorm in Transformer directly affect the magnitudes of tokens by standardizing their feature vectors. As demonstrated in Fig.~4 in the revision, by taking the perpendicular component $\Delta(X)[i,:]$ as the residual signal, the tokens are updated more in their tangent directions and less in their magnitudes. Intuitively, this prevents LN or RMSNorm from diminishing the effects that belief-attention has on token updates.

---

> > ### Comment · Reviewer_axrZ · 2025-11-28
> >
> > This is reasonable for me, thanks to the author's detailed explanation, and the content of the article is very enlightening for me, I would like to raise my rating to 6

---

### Official Review · Reviewer_XiS7 · 2025-11-06

**Soundness:** 2
**Presentation:** 3
**Contribution:** 2
**Rating:** 4
**Confidence:** 3

**Summary:**

This paper proposes Belief-Attention, a variant of Transformer attention inspired by ideas from the Primal-Dual Method of Multipliers (PDMM). Instead of directly using the standard residual connection from the attention output, Belief-Attention orthogonalizes this residual with respect to the value vectors $V$, encouraging updates that change direction rather than magnitude. A variant, Belief-Attention*, performs projection per head and introduces an additional learnable matrix $W_s$. Experiments on ViT (ImageNet), nano-GPT2 (OpenWebText), and CIFAR-10 show consistent but modest accuracy and loss improvements with limited computational overhead. The method is easy to implement and generalizes across both vision and language tasks.

**Strengths:**

The paper’s main strengths lie in its simplicity, generality, and clarity. The proposed modification is theoretically motivated and readily applicable across various architectures without requiring retraining or a structural overhaul. Empirical results show consistent gains in both vision and language tasks, and the discussion of limitations is transparent. The approach offers an intuitive geometric interpretation, emphasizing angular changes over magnitude, which may provide insights for understanding and stabilizing deep residual learning.

**Weaknesses:**

The main limitation is the lack of rigorous theoretical grounding. The PDMM analogy is suggestive but not formalized, and it remains unclear when or why orthogonal projections should improve optimization or generalization. The experiments, while broad, are shallow in statistical rigor—missing repetitions, variance reporting, and ablation details (e.g., projection strength, per-layer effects, normalization variants). Numerical stability and computational cost analyses are also underexplored. Finally, the paper does not discuss prior related works on orthogonality or residual reparameterization (e.g., ReZero, cosine similarity regularization), which would help situate the contribution.

**Questions:**

Could the authors explicitly map PDMM variables (residuals, multipliers, discrepancy) to quantities in the attention module (MH, V, X), clarifying what is retained and what is replaced in the Belief-Attention model?

Can the authors formalize or empirically justify conditions under which orthogonalizing residuals improves generalization or convergence, perhaps by connecting to Lipschitz continuity or residual scaling analyses?

How are LayerNorms positioned relative to the projection step, and are results sensitive to pre-LN versus post-LN configurations?

Would introducing a learnable projection strength (e.g., scaling factor $\gamma$) improve stability or adaptability?

How does the model handle near-zero norms in $V$ during the orthogonalization process? Are numerical safeguards (e.g., $\epsilon$-stabilization) applied, and what impact do they have?

Do the reported gains hold for larger-scale language models or long-context causal decoders?

Are the proposed modifications compatible with efficient attention variants such as FlashAttention or linear attention kernels?

---

> ### Author Response · Authors · 2025-11-25
>
> __* Could the authors explicitly map PDMM variables (residuals, multipliers, discrepancy) to quantities in the attention module (MH, V, X), clarifying what is retained and what is replaced in the Belief-Attention model?__
>
> Many thanks for the comments. We have revised the paper based on the comments. In the revision, we introduce quadratic-approximation based PDMM (QA-PDMM), which share a great structural similarity with a standard Transformer layer. Figure 1 in the revision visualizes the structural similarity between QA-PDMM and the Transformer layer. A new paragraph on page 4 explicitly explains the structural similarity in text.
>
> Basically, at each iteration $k$ in QA-PDMM, the estimate $x_i^{k+\frac{1}{2}}$ at node $i$ is computed by performing information aggregation from neighbors, followed by a skip-connection. $x_i^{k+1}$ is then computed at node $i$ with the local information $x_i^{k+\frac{1}{2}}$ and $x_i^{k-\frac{1}{2}}$, followed by another skip-connection. See the appendix of the revision for derivation of QA-PDMM from PDMM. From a high level point of view, the computation for $x_i^{k+\frac{1}{2}}$ in QA-PDMM corresponds to the attention layer for updating the $i$th token in Transformer and the computation for  $x_i^{k+1}$ corresponds to the FFN layer for updating the $i$th token again.
>
> __*  How are LayerNorms positioned relative to the projection step, and are results sensitive to pre-LN versus post-LN configurations?__
>
> Many thanks for the comments.  All the three open sources being exploited in the paper implemented either pre-LN or pre-RMSNorm by default. We didn't change pre-LN (or pre-RMSNorm) to post-LN (or post-RMSNorm) to favor our newly proposed belief-attention layer and its variant. According to the conclusions made in the paper  "On Layer Normalization in the Transformer Architecture" from 2020, Pre-LN is preferable in comparison to Post-LN, which exhibits fast training speeds and less hyper-parameter tuning. It is also known that the llama model from Meta also uses pre-RMSNorm instead of post-RMSNorm.
>
> __* It remains unclear why orthogonal projections should improve generalization.__
>
> Many thanks for the comments. We have added one more paragraph at the bottom of page 6 to provide an explanation why belief-attention leads to performance improvement.
>
> In brief, we note that layer normalization (LN) or RMSNorm in Transformer directly affect the magnitudes of tokens by standardizing their feature vectors. As demonstrated in Fig.~4 in the revision, by taking the perpendicular component $\Delta(X)[i,:]$ as the residual signal, the tokens are updated more in their tangent directions and less in their magnitudes. Intuitively, this prevents LN or RMSNorm from diminishing the effects that belief-attention has on token updates.
>
> __* The paper does not discuss prior related works on orthogonality or residual reparameterization (e.g., ReZero, cosine similarity regularization), which would help situate the contribution.__
>
> Many thanks for the comments. We have added a section in the revision, which briefly discussed the related literature including ReZero and  cosine similarity regularization.

---

> > ### Author Response · Authors · 2025-11-25
> >
> > __*  Would introducing a learnable projection strength (e.g., scaling factor ) improve stability or adaptability?__
> >
> > Many thanks for the comments. In general, there is no need to introduce a learnable projection strength for the residual signal. This is because $W_o$, the linear mapping in preparation for skip-connection, can merge with the learnable projection strength.
> >
> > On the other hand, thanks to the reviewer, we are now aware from the literature that the ReZero technique helps with training ResNet-type of neural networks. Due to time limitation in the rebuttal phase, we will investigate in the next step if the ReZero technique can be applied to belief-attention and its variant for performance improvement.
> >
> > __* How does the model handle near-zero norms in  during the orthogonalization process? Are numerical safeguards (e.g., -stabilization) applied, and what impact do they have?__
> >
> > Many thanks for the comments. We have revised our manuscript accordingly. To clarify, in our original experiments, there is no need to introduce a small positive value as a safeguard.  Based on your comments,  we have tested the performance recently when a safeguard was introduced when training nano-GPT with belief-attention. The performance is almost the same as the original one, which will be reflected in the final version.
> >
> > __* Do the reported gains hold for larger-scale language models or long-context causal decoders?__
> >
> > Many thanks for the comments. Due to limited GPU resources and limited time at the moment, we were not able to conduct experiments for larger-scale language models.
> >
> > On the other hand, we fully agree with the reviewers that it is of high interest to find out if the reported gains also hold larger-scale language models or not. We will perform the large scale experiments (e.g., training llama models) in  the next step by renting the GPU resources from either Google colab or from other platforms.
> >
> >
> > __* Are the proposed modifications compatible with efficient attention variants such as Flash-Attention or linear attention kernels?__
> >
> > Many thanks for the comments. In principle, the proposed belief-attention should be compatible with Flash-Attention. Because Flash-Attention is a technique to effectively handle the KV cache to save memory, which is orthogonal to the introduced operations in belief-attention.
> >
> > We note that this paper primarily focuses on standard attention layer. It would be very interesting to extend the belief-attention for linear attention kernels in a separate article.

---

### Author Response · Authors · 2025-11-25

The authors thank all four reviewers for their appreciation of the novelty and simplicity of our newly proposed belief-attention and its variant belief-attention$^{\ast}$. Notably, reviewer 5MXh states that __the geometric analysis of the orthogonal projection (directional vs magnitude changes) is sound and intuitive__. Reviewer XiS7 states that __the paper’s main strengths lie in its simplicity, generality, and clarity__. We would greatly appreciate the reviewers checking our detailed rebuttal responses and provide further guidance to improve the quality of our paper.

To briefly summarize, in the revision, we derive quadratic-approximation based PDMM (QA-PDMM) from PDMM (a distributed optimisation algorithm) for the purpose of establishing the connection between the update expressions of QA-PDMM and the Attention-FFN framework in a standard Transformer layer. We added a few paragraphs and a new figure to discuss the structural similarity of QA-PDMM and the Transformer layer. We hope the revision provides a strong and sound motivation for the design of belief-attention and its variant in Transformer from a distributed optimisation viewpoint.

We have also performed additional experiments (see the revised paper) as required by the four reviewers.  In particular, we evaluate Belief-ViT for image classification over CIFAR10 and CIAR100 (via the open-source https://github.com/aanna0701/SPT_LSA_ViT) and Belief-Llama (roughly 181M parameters)  for natural language processing (via the open-source https://github.com/hengjiUSTC/learn-llm/tree/main/pretrain) over FineWeb-Edu of size 10BT. Both Belief-ViT and Belief-Llama are obtained by replacing the standard attention layers with the proposed belief-attention$^{\ast}$.  It is found that the validation performance is considerably improved for the above tasks, which is consistent with our earlier experimental results.

We are now in the process of testing LLama (roughly 1B parameters) and Belief-Llama over FineWeb-Edu of size 10BT by using a single A6000 GPU. So far we have trained the models for 2250 iterations. Again we start to observe performance gain of Belief-Llama over Llama in terms of the validation loss.

---

### Note · Authors · 2026-03-22

I have read and agree with the venue's withdrawal policy on behalf of myself and my co-authors.

---

### Meta-Review · Area_Chair_mP4u · 2026-01-06

**Summary:**

Initial reviews are largely negative (2,4,4,4).

Reviewer XiS7 notes that the work lacks a theoretical grounding and questions whether the use of orthogonal projections should be beneficial.  Likewise, has questions regarding the experimental evaluation in various aspects.

Reviewer axrZ also has questions regarding the motivation for the core method and experimental evaluation while also noting the potential for increased computational cost.

Reviewer 5MXh likewise notes a relatively limited experimental evaluation and a somewhat heuristic presentation and development of the method.

Reviewer j7jm suggests that the models are not properly tuned and below standard performance benchmarks for the tested baseline architectures along with finding evidence to support the claim that the models can scale to be lacking (a sentiment also shared in other reviews).

**Reviewer Concerns:**

Overall, while the authors have made an earnest effort to address the reviewers concerns, I am unfortunately not convinced that many of the issues raised have been resolved.

For example, a majority of the reviewers raise concerns that the method is somewhat intuitive or heuristic without theoretical development.  In response the authors note modifications to the manuscript, but this would again appear to be largely intuitive or empirical argument rather than a clear formulation, derivation, etc as asked for by the reviewers.

Likewise, while the authors acknowledge the need for larger-scale language model experiments as requested by the reviewers, they were unfortunately unable to provide many due to computational resource limitations.

**Reviewer Scores:**

Two reviewers were able to engage in discussion with the authors before the discussion was closed.  Reviewer axrZ notes that they find the authors' response reasonable and raising their score.  Reviewer j7jm, however, largely discusses proper experimental protocols and settings with the authors without noting a change in opinion on the work or revision to their score.

Given the relatively low and unanimous initial reviews along with some of the potentially unresolved issues described above I unfortunately find it unlikely that the reviewers' scores would be revised significantly to reach a consensus for acceptance of the paper.  The authors appear to have received clear suggestions from the reviewers for how the manuscript can be improved and I would encourage them to take this feedback into account for future submissions.

---

### Decision · Program_Chairs · 2026-01-26

Reject